# The Nucleoside Analog GS-441524 Effectively Attenuates the In Vitro Replication of Multiple Lineages of Circulating Canine Distemper Viruses Isolated from Wild North American Carnivores

**DOI:** 10.3390/v17020150

**Published:** 2025-01-23

**Authors:** Arturo Oliver-Guimera, Brian G. Murphy, M. Kevin Keel

**Affiliations:** 1Department of Pathology, Microbiology, and Immunology, School of Veterinary Medicine, 4206 Vet Med 3A, University of California, Davis, CA 95616, USA; arturo.oliver@uchceu.es (A.O.-G.); bmurphy@ucdavis.edu (B.G.M.); 2Veterinary Pathology Service, School of Veterinary Medicine, Department of Animal Production and Health, Public Veterinary Health and Food Science and Technology, Universidad Cardenal Herrera-CEU, CEU Universities, 46115 Valencia, Spain

**Keywords:** antivirals, canine distemper, viral disease, dogs, vRNA, nucleoside analog

## Abstract

Canine distemper is a severe and lethal viral disease of dogs and wild carnivores with an urgent need for the identification of effective antiviral agents against canine distemper virus (CDV). We assessed multiple agents for their ability to block the replication of three different lineages of CDV isolated from wild carnivores in the United States. Six antiviral compounds were selected after preliminary experiments that excluded ribavirin, hesperidin and rutin: a protease inhibitor (nirmatrelvir), a polymerase inhibitor (favipiravir) and four nucleoside analogs (remdesivir, GS-441524, EIDD2801 and EIDD1931). Antiviral efficacy was determined by the attenuation of the cytopathic effect in a CDV-susceptible cell line and the inhibition of viral RNA replication. The nucleoside analog GS-441524 effectively blocked the replication of CDV at pharmacologically relevant concentrations. Four other antiviral agents inhibited CDV replication to a lesser degree (remdesivir, nirmatrelvir, EIDD2801 and EIDD1931). The replication of different viral lineages was differentially inhibited by the antivirals. Several of the nucleoside analogs have been safely used previously in carnivore species for the treatment of other viral diseases, suggesting that they may be promising candidates for the treatment of canine distemper in dogs. Our results emphasize the need to consider different viral lineages in the screening of antiviral compounds.

## 1. Introduction

Canine distemper is a highly contagious, severe and potentially lethal infectious disease of dogs and other carnivores caused by canine distemper virus (CDV) [1]. Despite the name, CDV does not exclusively infect dogs and has been determined to cause disease in a wide range of carnivores, including members of the *Felidae*, *Canidae*, *Mustelidae* and *Procyonidae* [2]. Although there are established vaccine protocols for dogs in the United States (US), CDV is still a relatively common disease due to its persistence in non-vaccinated dog populations and in wildlife, and possibly as a result of CDV vaccine failures [3]. Viral eradication is not a feasible short-term goal due to unvaccinated dog populations and susceptible wildlife species acting as reservoirs for CDV [4,5,6].

Canine distemper virus has a remarkably broad cell tropism and, as a result, can infect, replicate in and injure a large number of tissues, leading to a diversity of lesions and clinical disease manifestations. Infection generally starts in lymphatic tissues, resulting in lymphoid necrosis and depletion of lymphocytes, followed by immunosuppression. The virus then replicates in multiple epithelial tissues, potentially damaging the respiratory tract, alimentary tract, pancreas, liver, epididymis and testis, amongst other tissues [1,7,8]. Distemper can also infect the brain, bone and ocular tissues, resulting in associated lesions [9,10]. A rare manifestation of the disease is commonly referred to as “old dog encephalitis”, which is suspected to be the result of persistent infection with a replication-defective virus [9].

Canine distemper virus is a paramyxovirus of the genus *Morbillivirus*. As an RNA virus, CDV has a relatively high mutation rate. The gene encoding the surface glycoprotein hemagglutinin (H) is one of the most variable CDV genes, and as a result, the sequence of hemagglutinin is typically utilized to determine and define viral lineages or genotypes (defined by sequence divergence of 5% or more) [11,12,13]. At least twenty different CDV lineages have been reported [14,15,16], with at least five of them (America-2, America-3, America-4, America-5 and Rhode Island-like) currently circulating in the United States [16,17,18]. The effects of viral sequence variation on the clinical signs and tissue lesions associated with CDV infection are presently unknown, but specific viral mutations have been associated with changes in host range and pathogenicity [19,20]. There is no currently effective Food and Drug Administration-approved treatment for canine distemper in the US or similar approvals in other countries. Management of infected individuals is focused on supportive care [3,21]. The identification of a safe and effective antiviral therapy would be highly beneficial for the canines as well as other susceptible mammalian species.

A variety of nucleoside analog drugs have been demonstrated to be effective in the treatment of several RNA infections. These compounds bind to the conserved active site of the viral RNA-dependent RNA polymerase (RdRp) in their triphosphate form, competing with the actual nucleoside substrates and resulting in early chain termination of the nascent viral RNA strand [22,23]. The viral RdRp is required for the replication of RNA viruses, because this enzymatic activity is absent in the mammalian cells they infect. Nucleoside analog drugs block this common molecular pathway of RNA viruses, giving them the potential to act as broad-spectrum antiviral agents. Three analogs stand out for their broad-spectrum action against RNA viruses: remdesivir, ribavirin and favipiravir [22]. Remdesivir, a phosphoramidate prodrug that is metabolized into an adenosine analog [24,25], has demonstrated antiviral activity against several families of RNA viruses, including *Filoviridae*, *Coronaviridae*, *Paramyxoviridae* and *Pneumoviridae* [26,27,28,29]. The active metabolite of remdesivir, GS-441524, has been successfully used to successfully treat feline infectious peritonitis in vivo, a disease with no previous therapeutic options [28,29,30,31,32]. Ribavirin, a broad-spectrum purine nucleoside analog, is effective against filoviruses, paramyxoviruses [33] and hepatitis E virus [34]. Favipiravir, which also has demonstrated direct inhibitory activity of the RNA-dependent RNA polymerase, is considered an effective antiviral compound against rabies, influenza, Ebola and several paramyxoviruses [35,36,37].

The flavonoids have also been reported to have antiviral activity for many viruses, including influenza and dengue viruses [38,39,40]. The drugs in this group are plant metabolites that include compounds such as rutin and hesperidin. Although the specific mechanism of action is not completely understood, they are suspected to inhibit the viral polymerase [41] or to interact with viral surface glycoproteins (such as hemagglutinin) to prevent cell–virus attachment interactions [42,43].

Several antiviral compounds are known to inhibit the replication of other paramyxoviruses or even morbilliviruses [44,45,46]. Although only a few of these agents have been tested against CDV, the similarity in protein composition and replication strategies of CDV with the other members of their family suggests that they might be good candidates for distemper treatment as well. The nucleoside analogs remdesivir and GS-441524 have been shown to effectively restrict the replication of another morbillivirus, measles virus [44,47]. Ribavirin can reduce the cytopathic effect of the measles virus [48], and favipiravir has an antiviral effect against the morbillivirus peste des petits ruminants virus [49]. EIDD1931 and its prodrug EIDD2801 are efficient inhibitors of respiratory syncytial virus [50,51]. Some flavonoids and nucleoside analogs have also decreased viral replication in cells that were infected with CDV [43,47,52,53].

Because some of these compounds have been utilized for the treatment of other animal diseases, including other canine diseases, their toxicity and pharmacodynamic properties have previously been defined. Therefore, repurposing these compounds as distemper treatments may be feasible. Favipiravir has been used as an effective treatment for influenza in mouse models [35,54], and the toxic dose in dogs has been determined to be very high [55]. The flavonoid rutin is the recommended treatment for chylothorax in dogs [56,57], and an additional flavonoid, hesperidin, is a recommended pulp-capping agent used to seal dental pulp cavities [58].

Clinical trials for new drugs are necessary to establish the efficacy and safety of new drugs; however, they are expensive and require the use of live animals, which requires complex logistics and specialized training [59,60]. To maximize the success of clinical trials, it is crucial to identify excellent candidates that can minimize the number of animals needed and the overall cost of drug testing [61,62]. Additionally, drugs become more robust candidates for the treatment of specific diseases when multiple independent sources support their efficacy.

In this study, we determined the efficacy of a slate of antiviral agents for the in vitro treatment of canine distemper virus using a multi-modal screening assay.

## 2. Materials and Methods

### 2.1. Cell Lines and CDV Isolates

The cell line used for this experiment was Vero cells (derived from the kidney of an African green monkey) that were transfected to express dogSLAM, a well-known receptor for CDV. Vero dogSLAM cells were grown on Dulbecco’s Modified Eagle Medium (DMEM, Gibco, Buffalo, NY, USA) supplemented with 5% of fetal bovine serum (FBS, GenClone, EL Cajon, CA, USA) and 1% antibiotic-antimycotic combination (anti-anti 100×, Gibco, Buffalo, NY, USA). They were incubated at 37 °C in an atmosphere with 5% CO_2_.

The CDV isolates for this study were obtained from three clinically affected wild carnivores in the USA, including two gray foxes and one raccoon. The isolates were named GA-299 (from Louisiana, donated by Nicole Nemeth from the University of Georgia), CA-914 (from California) and NH1 (from New Hampshire, donated by David Needle from the University of New Hampshire). Their lineages and other details are indicated in Table 1. To isolate the viruses, filtered tissue homogenates from these animals were incubated with Vero dogSLAM cells until cytopathic effect (cell detachment, syncytia formation and cell death) was observed. At this time, the culture supernatant was collected and used to infect new Vero dogSLAM cells. After three passages, workable concentrations (>10^3^ plaque-forming units (PFUs)/mL) were achieved, and supernatants were stored at −80 °C.

Total RNA was extracted from 140 μL of the cell culture supernatant virus using a QIAamp viral RNA Mini kit (QIAGEN, Germantown, MD, USA) according to manufacturer instructions. The presence of CDV-RNA was confirmed using standard one-step RT-PCR (One-step RT-PCR, QIAGEN, Germantown, MD, USA) in a C1000 Touch Thermal Cycler (Bio-Rad, Hercules, CA, USA). Previous CDV isolates were used as positive controls, and the negative control consisted of the master mix with nuclease-free water instead of a viral template. All primers were custom-made. The forward primer was 5′-ATGAAACGATCCCCAGGG-3′, the reverse primer was 5′-ACTGATGTAACACTGGTCT-3′, and they targeted an 880 nucleotide-long segment in the CDV nucleocapsid gene. Conditions were 30 min at 50 °C, 15 min at 95 °C, 40 cycles of 30 s at 94 °C, 30 s at 50 °C and 1:30 min at 72 °C and a final step of 10 min at 77 °C. Amplicons were run in 1% agarose electrophoresis gel, and the presence of CDV-RNA was confirmed by the identification of a band of expected size in comparison to a positive control.

### 2.2. Antivirals

The antivirals used in this study and their chemical details are summarized in Table 2. All antivirals used in this study were tested at concentrations of 5, 1.5 and 1.25 μM unless stated otherwise. Drugs were diluted in dimethyl sulfoxide to concentrations of 100 μM and then in cell culture media to working concentrations. The list of drugs tested includes favipiravir; nirmatrelvir; the nucleoside analogs remdesivir, ribavirin, GS-441524, EIDD2801 and EIDD1931 (Natural Micron Pharm Tech, Taian City, Shandong, China); and two flavonoids, rutin (Rutin hydrate, Sigma, Burlington, MA, USA) and hesperidin (Sigma Aldrich, Burlington, MA, USA).

### 2.3. Antiviral Screening

The viral plaque assay was used as described previously [27]. Briefly, 96-well plates (Corning) were seeded with 10^4^ Vero dogSLAM cells and infected with CDV at confluency. Six-well replicates of infected cells were treated with antiviral compounds and compared to controls run in parallel. The controls included untreated infected cells, uninfected treated cells and wells with virus but no cells. Cells were infected at the point of cell confluency, with 200 µL of media containing CDV at a multiplicity of infection (MOI) of 0.04 for one hour. Antiviral treatments were then added at a defined final concentration for each well. Tissue culture plates were incubated at 37 °C in an atmosphere of 5% CO_2_ for 3 additional days. A total of 100 µL of supernatant was collected for RNA extraction. The adherent cells were then fixed with buffered 10% formalin and stained with 0.5% crystal violet (Fisher chemical, Pittsburgh, PA, USA). Virus-associated cell death was evaluated by partial to complete loss of the monolayer when compared to mock-infected control cells as quantified by light absorbance. The absorbance for each well was measured at 620 nm using an ELISA plate reader (FilterMax F3 [Molecular Devices, San Jose, CA, USA] and Softmax Pro [Molecular Devices, San Jose, CA, USA]). The mean absorbance and standard deviation of the mean measurements of the six-well replicates were used for statistical analyses. For agents that demonstrated antiviral efficacy (reduction in virus-associated cell death), the half-maximal effective concentration (EC_50_) was determined by plotting a nonlinear regression equation (i.e., a dose-response curve) using GraphPad Prism version 9.4.1 for MacOS (GraphPad Software, San Diego, CA, USA).

### 2.4. Quantitative RT-PCR

One-step qRT-PCR was performed using AgPath-ID One-step RT-PCR Kit (Applied Biosystems, Waltham, MA, USA) in a 7500 Fast and 7500 Real-Time PCR System (Applied Biosystems). PCR conditions were as follows: 10 min at 42 °C, 6 min at 95 °C and 40 cycles of 3 s at 95 °C and 30 s at 60 °C. Primers targeting an 88 nucleotide long fragment of the nucleocapsid gene were used. The forward primer was 5′-AGCTAGTTTCATCTTAACTATCAAATT-3′, the reverse primer was 5′-TTAACTCTCCAGAAAACTCATGC-3′, and the probe was 5′-/5-FAM/ACCCAAGAGCCGGATACATAGTTTCAATGC/36-TAMSp/-3′. The efficacy of treatments was calculated as Δ(40 − Ct), representing the 40 − Ct difference between each measurement and the average Ct for untreated cells for the same viral isolate. For statistical analysis, Δ(40 − Ct) values were compared, as well as 2^Δ(40 − Ct)^, to accurately represent viral RNA copy reduction. Therefore, a value of 1 (Δ = 0) indicated no difference between treated and untreated cells.

### 2.5. Statistical Analysis

The normality of data was tested using the Wilcoxon–Shapiro test. A Two-way ANOVA, or its non-parametric equivalent, was used to detect differences in absorbance and Ct (viral RNA loads) between different qRT-PCR products. A pairwise comparison was performed using Tukey’s test. To calculate antiviral efficacy for viral replication, one sample *t*-test was used to compare Δ(40 − Ct) with 0 (no effect). All *p* values ≤ 0.05 were considered statistically significant. The analyses were performed using GraphPad Prism version 9.4.1 for MacOS (GraphPad Software, San Diego, CA, USA).

## 3. Results

### 3.1. Antivirals and Cytopathic Effect

Cell culture monolayers infected with CDV developed visually apparent viral plaques and attenuation of the cellular monolayer (Figure 1). The treated, mock-infected cells did not demonstrate plaques even at the highest concentration tested for every compound. The only exception was a slight decrease in cell density with 5 µM favipiravir treatment (*t*-test, *p*-value < 0.05).

The cytopathic effect (CPE) of the America-4 isolate, GA-299, was reduced relative to the other viral isolates, even at the same concentrations of virus. Since cell death was not visually quantifiable even in the absence of treatment, the protective effects of antiviral drugs could not be tested for this isolate, although changes in viral RNA production were quantified since they were measurable with qRT-PCR.

Five µM of GS-441524 was the most effective treatment at preventing virus-associated cell death (*p* < 0.0001, Tukey’s test) for both CA-914 and NH1. At this concentration, GS-441524 inhibited cell death to levels comparable to those of mock-infected cells (Figure 2). Remdesivir, also at 5 µM, was associated with a degree of protection relative to untreated infected cells (*p* < 0.001, Tukey’s test), and this protection was higher than all the other drugs except GS-441524 (*p* < 0.01, Tukey’s test). Both GS-441524 and remdesivir-treated wells also had significantly lower CPE than untreated wells (*p* < 0.01, Tukey’s test). All of the other antiviral agents had little to no effect on preventing CPE (Figure 2 and Figure 3). We determined the EC_50_ for GS-441524 in CA-914 to be 2.72 μM (Figure 3).

### 3.2. Antivirals and CDV Replication

At a dosage of 5 µM, nirmatrelvir, remdesivir, GS-441524, EIDD2801 and EIDD1931 all significantly reduced the amount of viral RNA produced for two viral strains, GA-299 (*p* < 0.01, one sample *t*-test) and CA-914 (*p* < 0.001, one sample *t*-test). For NH1, the only effective treatment was GS-441524 when compared to untreated infected cells (*p* < 0.001, one sample *t*-test). The treatment with GS-441524 had the largest effect at reducing viral RNA production in all viral strains with up to 4 logarithmic reductions in viral RNA loads for all viral strains (*p* < 0.0001, two-way ANOVA, Figure 4 and Figure 5). There were no differences in viral RNA reduction when comparing nirmatrelvir, remdesivir, EIDD2801 and EIDD1931.

Hesperidin and rutin did not demonstrate any efficacy in preventing cell death or viral replication at any tested concentration up to 100 μM. The same was true for ribavirin at concentrations up to 20 μM.

## 4. Discussion

In this study, we determined the efficacy of several antiviral compounds with a reported ability to prevent viral replication and cell death. The nucleoside analog GS-441524 demonstrated promising antiviral results, suggesting that this agent is a strong candidate for further antiviral assessment in CDV-infected dogs or other mammals.

The importance of developing an efficient treatment for canine distemper is clear from the number of studies attempting to identify effective compounds for reducing the viral activity of CDV in vitro [63,64,65]. In spite of this, there is no current effective antiviral treatment for distemper. The identification of multiple effective antiviral treatments with different mechanisms of action is important to ensure the availability of therapeutic options under the emergence of resistant viral strains, as can happen in RNA viruses such as SARS-CoV-2 [66].

We identified no drug-associated cytotoxicity at 5 μM for all of the compounds except for favipiravir. This concentration is higher than the EC_50_ for any related viruses in all the drugs. The most effective drug in the study was the nucleoside analog GS-441524, a metabolite of the prodrug remdesivir [44,67]. GS-441524 has good oral bioavailability, and it seems to lack adverse reactions even at high doses, with a 50% cytotoxic concentration that is consistently above 50 μM in dogs [68]. This medication is well-tolerated in dogs orally at a maximum feasible dose of 2000 mg/kg [69]. In comparison, the equivalent dose of GS-441524 to potentially treat humans for COVID-19 has been calculated to be less than 4 mg/kg, and concentrations of 13 mg/kg have been administered to a healthy human without major adverse reactions [68,69,70]. In cats, GS-441524 has been proven effective experimentally in subcutaneous doses of 4 mg/kg [30,31], and it has been administered orally at concentrations of up to 25 mg/kg without major complications [28]. For the measles virus, a morbillivirus related to CDV, the EC_50_ for GS-441524 has been calculated to be 0.58 μM in Vero cells [44]. In our study, GS-441524 EC_50_ was 3.95 μM but was determined using cell protection rather than a fluorescent reporter.

Remdesivir, the prodrug of GS-441524, also demonstrated a protective effect on cells infected with CDV, although at a smaller magnitude. The EC_50_ of remdesivir for measles virus has been determined to range from 1 to 4.97 μM, depending on the study [44,47]. Despite the protective effect of remdesivir identified in our study (Figure 2), higher concentrations than the ones tested in this study would be needed to calculate an accurate EC_50_ for remdesivir (Figure 3). EIDD1931 and its prodrug EIDD2801 have previously been demonstrated to be efficacious for the treatment of RNA viruses such as coronaviruses, orthomyxovirus and paramyxoviruses, and their EC_50_ values range from 0.006 μM to 3.7 μM. However, these agents did not demonstrate an overt protective effect in our study [27,50,51].

Our qRT-PCR assay targeted the viral nucleocapsid gene, which is the first gene to be transcribed and replicated during CDV infection. In the case of GA-299, viral RNA production was measurable despite no visible cytopathic effect. It is possible that viral RNA replication occurred in the absence of a cytopathic effect. The measurement of viral RNA reduction proved to be a more sensitive method of detecting antiviral efficacy relative to visual evidence of cell protection. It is possible that their antiviral effects were not enough to prevent cell death or that they just reduced viral RNA production but not infective particle production. In accordance with the biological assay, GS-441524 was the most effective treatment in all viral isolates. Nirmatrelvir, remdesivir, EIDD2801 and EIDD1931 all had similar inhibitory effects except for the viral isolate NH1, for which they did not reduce viral RNA production. The isolate CA-914 appeared to be more sensitive to antiviral treatments. These results highlight the importance of using diverse viral isolates to test the efficacy of antiviral treatments against variant virus strains.

Nirmatrelvir is an effective and approved treatment for COVID-19, and it acts by inhibiting a specific protease of coronaviruses [71,72]. As such, it was not expected to have an antiviral effect on other RNA viruses that lack this protease, such as CDV. However, it did reduce viral RNA production as measured by qPCR in our experiments. It is possible that nirmatrelvir operates in different, still undescribed ways to block viral replication. The agents EIDD1931 and EIDD2801 were also not effective at preventing cell death, but they reduced viral RNA loads. These broad-spectrum ribonucleoside inhibitors have been tested in RNA viruses, but the closest relative of CDV for which they have shown an antiviral effect is the respiratory syncytial virus. Respiratory syncytial virus belongs to the same order as CDV, but it is in a different family, *Pneumoviridae* [73].

Despite previous reports of the effectiveness of rutin [43], hesperidin [43], favipiravir [53] or ribavirin [52,74] in the reduction of CDV replication in vitro, we did not see any protective effects. There are many instances in which our model differs from that of previous studies. Most of the previous studies focused on viral replication rather than pathogenic effects. Additionally, these antiviral drugs were tested in Vero cell cultures that were not modified to express CDV-specific receptors. In the case of rutin and hesperidin, cells were infected with Rockborn CDV, an ancient lineage that is only distantly related to the current circulating lineages [75]. Similarly, investigations of the efficacy of ribavirin for limiting CDV infection used the vaccine strain Onderstepoort, which belongs to the America-1 lineage [75]. For favipiravir, the specific lineage of the virus used was not disclosed, but it probably belonged to the Asian clade. The CDV isolates used in our study belong to America-3 (CA-914), America-4 (GA-299) and Rhode Island-like (NH-1), all of which currently cause disease in wildlife and dogs in the USA [18]. It is difficult to attribute the differences in outcomes between studies only to the use of different lineages of CDV. However, lineages differ mostly in the genotype of surface glycoprotein hemagglutinin, and one of the proposed antiviral mechanisms for flavonoids involves interaction with viral envelope glycoproteins, which would provide a reasonable explanation, at least in this group of drugs [42].

It is also noteworthy that differences in the pathogenesis of viral strains were evident in our study. Despite equivalent initial infectious doses, GA-299 had a markedly lower cytopathic effect that made the effects of antiviral compounds less evident. The explanation for this difference is not clear. One possibility could be the organ of origin of the isolate, but both GA-299 and CA-914 were isolated from brains and showed drastically different results. The GA-299 differed from the other two in the host species (raccoon vs. gray fox). Moreover, clinical records of the cases suggest that the disease presentation of the raccoon was milder than that of the gray foxes. Further studies are needed to understand the factors that influence distemper severity. However, this difference in cytopathic effect underscores the importance of using several diverse viral isolates when testing the effectiveness of antiviral treatments.

Our study showed that several nucleoside analogs effectively reduce CDV-associated cytological pathogenesis and replication in vitro, even in different CDV isolates, demonstrating that they are promising antiviral candidates to advance into preclinical studies for the treatment of distemper. Further research on their capacity to treat distemper in vivo is warranted.

## Figures and Tables

**Figure 1 viruses-17-00150-f001:**
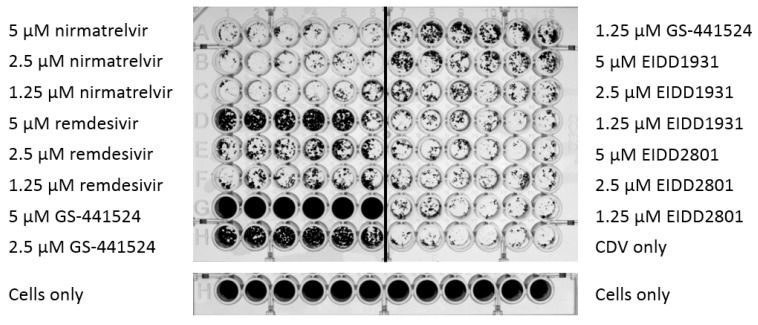
Example plate of antiviral screening. Vero dog-SLAM cells were infected with isolate NH-1 of canine distemper virus (CDV) both with and without different concentrations of the antiviral candidates indicated. Plaques and monolayer confluency were measured by staining the monolayers with crystal violet and measuring the absorbance at 620 nm. GS-441524 and remdesivir demonstrated antiviral effects, but there was no significant difference between wells treated with nirmatrelvir, EIDD1931 and EIDD2801 in comparison to the untreated CDV-infected wells.

**Figure 2 viruses-17-00150-f002:**
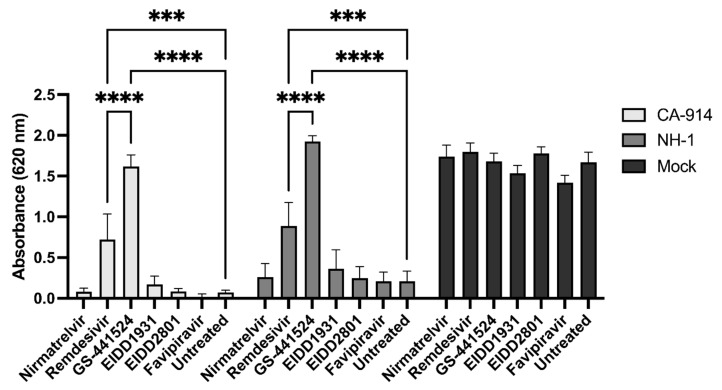
Effectiveness of antivirals at reducing cell death at 5 µM. Monolayer integrity was measured by absorbance at 620 nm after staining with crystal violet. GS-441524 and remdesivir had a significant increase in absorbance that correlated with higher cell survival than any of the other treatments (statistical significance not shown, *p*-value < 0.01). GS-441524 had increased absorbance compared to remdesivir for both NH-1 and CA-914 (****, *p*-value ≤ 0.0001). Remdesivir had increased absorbance compared to untreated cells for both NH-1 and CA-914 (***, *p*-value ≤ 0.001).

**Figure 3 viruses-17-00150-f003:**
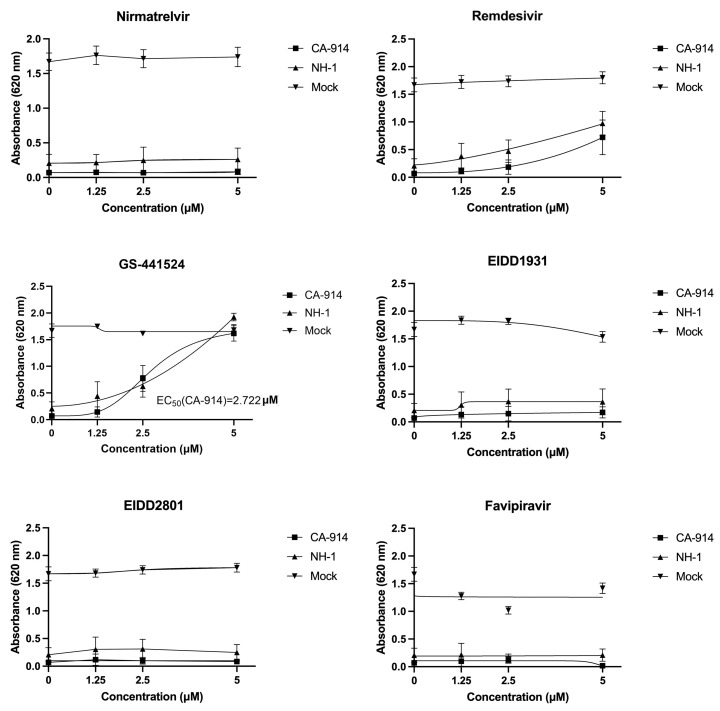
Effectiveness of antivirals at reducing cell death by concentration of antiviral treatment. Monolayer integrity was measured by absorbance at 620 nm after staining with crystal violet. Remdesivir and GS-441524 had a significant increase in absorbance that correlated with higher concentrations of the antiviral drug (*p*-value < 0.05).

**Figure 4 viruses-17-00150-f004:**
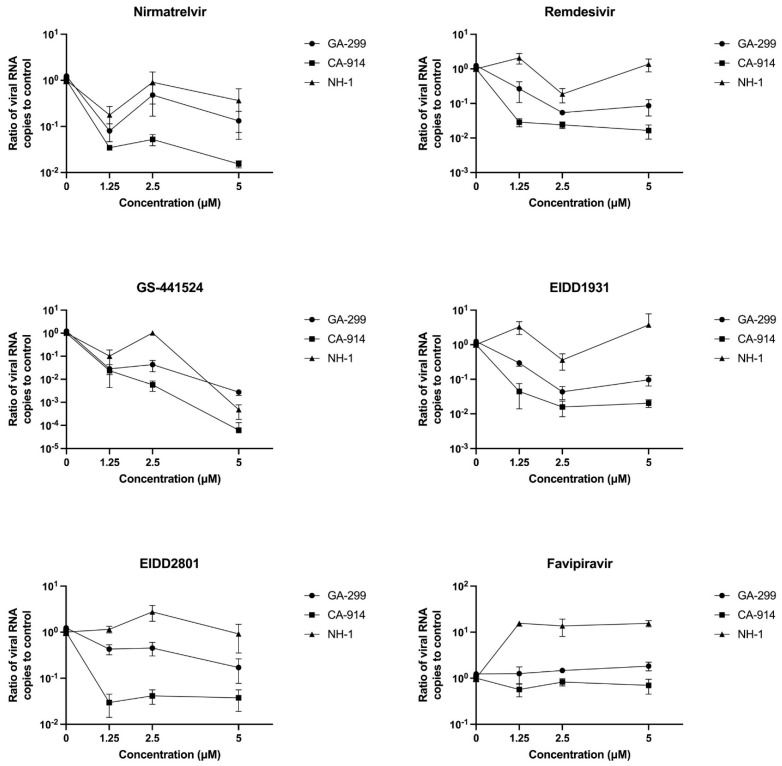
Effectiveness of antivirals at reducing viral RNA load by concentration of antiviral treatment. RNA reduction was calculated as RNA load reduction compared to untreated infected controls.

**Figure 5 viruses-17-00150-f005:**
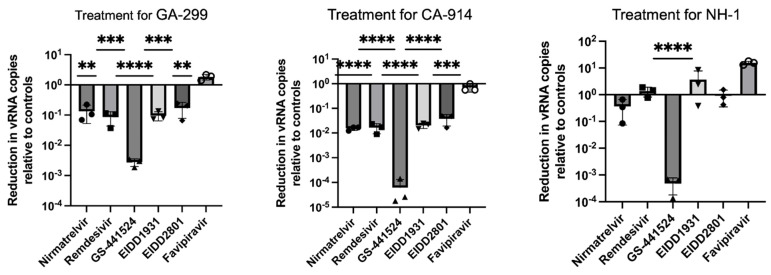
Effectiveness of antivirals at reducing viral RNA load at 5 µM. RNA reduction was calculated as 2^Δ(40 − Ct)^ for reduction with respect to untreated infected controls. All treatments but faviparavir, especially GS-441524, significantly reduced viral load in GA-299 (**, *p*-value ≤ 0.01; ***, *p*-value ≤ 0.001, ****, *p*-value ≤ 0.0001) and CA-914 ((***, *p*-value ≤ 0.001; ****, *p*-value ≤ 0.0001), while only GS-441524 reduced viral RNA load in cells infected with NH1 (****, *p*-value ≤ 0.0001).

**Table 1 viruses-17-00150-t001:** Summary of isolates used in this study. CA-914 was collected in our institution, while GA-299 and NH1 were donated by Nicole Nemeth from the University of Georgia and David Needle from the University of New Hampshire, respectively.

Isolate	Lineage	Origin Species	Organ	Geographical Origin
GA-299	America-4	(Raccoon, *Procyon lotor*)	Brain	Louisiana
CA-914	America-3	(Gray fox, *Urocyon cinereoargenteus*)	Brain	California
NH1	Rhode Island	(Gray fox, *Urocyon cinereoargenteus*)	Lung	New Hampshire

**Table 2 viruses-17-00150-t002:** Summary of the antiviral compounds used in this study.

Compound	Mechanism of Action	Chemical Formula	Structural Formula
EIDD1931	Nucleoside analog (cytosine)	C_9_H_13_N_3_O_6_	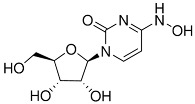
EIDD2801	Nucleoside analog (cytosine)	C_13_H_19_N_3_O_7_	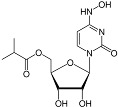
Favipiravir	Polymerase inhibitor	C_5_H_4_FN_3_O_2_	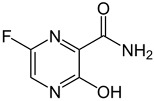
GS-441524	Nucleoside analog (adenosine)	C_12_H_13_N_5_O_4_	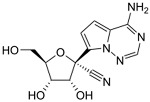
Hesperidin	Flavonoid (specific mechanism unknown)	C_28_H_34_O_15_	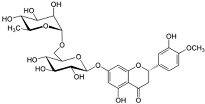
Nirmatrelvir	Protease inhibitor	C_23_H_32_F_3_N_5_O_4_	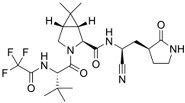
Remdesivir	Nucleoside analog (adenosine)	C_27_H_35_N_6_O_8_P	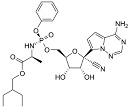
Ribavirin	Nucleoside analogs (purine)	C_8_H_12_N_4_O_5_	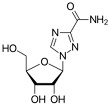
Rutin	Flavonoid (specific mechanism unknown)	C_27_H_30_O_16_	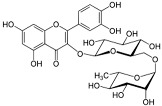

## Data Availability

The original contributions presented in this study are included in the article/Appendix A. Further inquiries can be directed to the corresponding author.

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
