# Peer review of "The Nucleoside Analog GS-441524 Effectively Attenuates the In Vitro Replication of Multiple Lineages of Circulating Canine Distemper Viruses Isolated from Wild North American Carnivores"

_viruses, 2025, doi:10.3390/v17020150_

Round 1

Reviewer 1 Report

Comments and Suggestions for Authors

In the manuscript « The nucleoside analog GS-441524 effectively attenuates the in vitro replication of multiple lineages of circulating canine distemper viruses isolated from wild North American carnivores », Oliver-Guimera et al. tested different compounds to inhibit three CDV isolates.

Here are my comments :

1) In the introduction the authors should add a paragraph about the old dog encephalitis caused by CDV infection.

2) In the materials and methods the authors should detail how they amplified the viruses after the isolation from the tissue homogenates. What titers are expected and how many passages are necessary to obtain workable titers? 

3) Lines 145 to 149 : it would be great to add the reference for all the antivirals.

4) Lines 172 to 183 : how are the viral copies calculated? Are they standardized with a house keeping gene? 

5) In figure 1 which CDV strain has been used? 

6) Lines 205 to 209 and overall in the manuscript, the sequencing of the isolates would really help to interpret the data and to determine the differences between the isolates.  

7) In figures 4 and 5 it would be great to have some raw data avec the CT and the basis viral RNA copies in order to have an idea about the level of the infection and the inhibition.

8) In the discussion the authors could discuss the difference of the lung and brain origin from the isolates and how is it important for the antivirals efficacy.

Author Response

The authors appreciate the comments and suggestions from reviewer 1

1) In the introduction the authors should add a paragraph about the old dog encephalitis caused by CDV infection.

Response 1: Thank you for your suggestion. We have added a sentence to mention old dog encephalitis (lines 46-48).

2) In the materials and methods the authors should detail how they amplified the viruses after the isolation from the tissue homogenates. What titers are expected and how many passages are necessary to obtain workable titers? 

Response 2: the suggested information has been added in the Cell lines and CDV isolates section of Materials and Methods (lines 130-132).

3) Lines 145 to 149: it would be great to add the reference for all the antivirals.

Response 3: the authors would like clarification if this is an important point. What references are you looking for? the publication references or more data on the source of the antiviral drugs?

4) Lines 172 to 183: how are the viral copies calculated? Are they standardized with a house keeping gene? 

Response 4: we considered that calculating viral copies using a house keeping gene from the host cells was not a good idea. Since CDV is a cytolytic virus, host cells gene expression will decrease with cell death. Therefore, we calculated viral load changes using changes in Ct: Δ(40-Ct) or 2Δ(40-Ct) depending on the analysis. This is explained at the end of the Quantitative RT-PCR section of Materials and Methods.

5) In figure 1 which CDV strain has been used? 

Response 5: Thank you for pointing out we did not mention which CDV was used. It was NH-1 and it has now been included in the figure legend (lines 208-209).

6) Lines 205 to 209 and overall in the manuscript, the sequencing of the isolates would really help to interpret the data and to determine the differences between the isolates.

Response 6: We agree with this comment. Unfortunately, full genome sequencing of the viral isolates is still being performed and we could not complete in time for the CDV special issue. We expect to be able to publish them soon.

7) In figures 4 and 5 it would be great to have some raw data avec the CT and the basis viral RNA copies in order to have an idea about the level of the infection and the inhibition.

Response 7: We have included a supplementary table including the raw data of the Cts resulting from qRT-PCR.

8) In the discussion the authors could discuss the difference of the lung and brain origin from the isolates and how is it important for the antiviral’s efficacy.

Response 8: We have added a brief discussion about the possible explanations for the differences in the results among viral isolates (lines 345-352).

Reviewer 2 Report

Comments and Suggestions for Authors

In this manuscript, the authors present an in vitro experiment to evaluate the effectiveness of various antiviral compounds, representing different classes, in inhibiting the infection and replication of three distemper virus field strains in cell culture. The experimental design is sound, and the results are clear and promising. However, some minor revisions are required.

Minor changes:

Lane 69-72: Remdesivir has also been clinically successfully applied for the treatment of feline infectious peritonitis. Please cite a couple of studies.

Lane 72-74: There are many more recent publications demonstrating the efficacy of GS in clinical applications. Please cite.

Lane 115: what kind of cells are Vero cells? Please explain for the reader.

Lane 132: specify: extracted from virus-containing cell culture supernatant, volume?

Lane 133: “CDV-RNA” it’s what you detect by RT-PCR

Lane 145: what’s the rationale of using these concentrations?

Lane 156: I imagine that the cells and not the wells were infected..

Lane 172: Chapter “Quantitative RT-PCR”.: please specify in this chapter which CDV gene is amplified. Has this RT-qPCR been validated in another publication? Do you have data on specificity, sensitivity, linearity and dynamic range?

Fig: 3(written figure 2): the figure seems to be twice the same or what is the difference???

Results: it would be of help to use subtitles for the results

The authors can update reference 67, the link is still valid

Lane 302: change to variant virus strains

Author Response

The authors appreciate the comments and suggestions from reviewer 2

Comment 1. Lane 69-72: Remdesivir has also been clinically successfully applied for the treatment of feline infectious peritonitis. Please cite a couple of studies.

Response 1: Thank you for your suggestion. The authors meant to reference the success of remdesivir treating Feline infectious peritonitis when mentioning the Coronaviridae family and with reference 28. For emphasis, we have included another reference, number 29 (line 74).

Comment 2. Lane 72-74: There are many more recent publications demonstrating the efficacy of GS in clinical applications. Please cite.

Response 2: references number 29 and 32 were added about the efficacy of GS-441524 to treat PIF in clinical applications (line 76).

Comment 3. Lane 115: what kind of cells are Vero cells? Please explain for the reader.

Response 3: We have included the origin of Vero cells in the methods section (lines 117-118).

Comment 4. Lane 132: specify: extracted from virus-containing cell culture supernatant, volume?

Response 4: We have specified the sample used for RNA extraction (cell-culture supernatant) and the volume (140 μl) (line 137).

Comment 5. Lane 133: “CDV-RNA” it’s what you detect by RT-PCR

Response 5: Thank you for pointing this out. The sentence has been corrected (line 139 and 147).

Comment 6. Lane 145: what’s the rationale of using these concentrations?

Response 6: this range corresponded with effective concentrations for the analyzed compounds in previous studies. Moreover, preliminary studies determined this range for most compounds to show antiviral effect.

Comment 7. Lane 156: I imagine that the cells and not the wells were infected.

Response 7: You are correct. We have corrected the phrasing in this line (line 163).

Comment 8. Lane 172: Chapter “Quantitative RT-PCR”.: please specify in this chapter which CDV gene is amplified. Has this RT-qPCR been validated in another publication? Do you have data on specificity, sensitivity, linearity and dynamic range?

Response 8: Thank you for pointing out that the gene information was missing. It has been added (lines 184-185).

Comment 9. Fig: 3(written figure 2): the figure seems to be twice the same or what is the difference???

Response 9: Duplicated image has been eliminated and the name update to figure 3

Comment 10. Results: it would be of help to use subtitles for the results

Response 10: Two subtitles have been added: Antivirals and cytopathic effect and Antivirals and viral replication (lines 201 and 240).

Comment 11. The authors can update reference 67, the link is still valid

Response 11: The reference has been updated

Comment 12. Lane 302: change to variant virus strains

Response 12: The sentence has been changed (line 313).

Reviewer 3 Report

Comments and Suggestions for Authors

This work describes the activity of several nucleoside analogues and flavonoid derivatives against several isolates of canine distemper virus (CDV), which infects domestic animals and wild carnivores. It was shown that remdesivir and its analog GS-441524 effectively decreased reproduction of two CDV strains. Additionally, these antiviral drugs were tested in modified Vero cell cultures (VeroDOGSlam) for more objective antiviral and cell toxicity studies. Based on analysis of antiviral activity on several isolates a prognosis of therapeutic potency to block distribution of CDV can be made. This work can be recommended for publication.

Several questions remain:

1) How could you explain the better antiviral activity of GS-441524 compared to its phosphorylated analog Remdesivir? The reverse result seems to be more expected.

2) Illustration of nucleoside structures could substantially increase a quality of the manuscript and make it more readable.

3) Please, correct Remdesvir in schemes and figures (see Fig. 2 for example).

4) Page 15, line 509, Ref.63 (2021 - bold).

Author Response

The authors appreciate the comments and suggestions from reviewer 3.

1) How could you explain the better antiviral activity of GS-441524 compared to its phosphorylated analog Remdesivir? The reverse result seems to be more expected.

Response 1: Remdesivir is a phosphoramidated version of GS-441524 (https://opendata.ncats.nih.gov/covid19/GS-441524). According to the published literature, pharmacokinetic studies reveal that most of the remdesivir administered to patients transforms into GS-441524 which then enters the cells and eventually becomes its active triphosphorilated form (https://pmc.ncbi.nlm.nih.gov/articles/PMC7315846/). GS-441524 needs one fewer step to performs its action, since it only needs the action of kinases.

2) Illustration of nucleoside structures could substantially increase a quality of the manuscript and make it more readable.

Response 2: Thank you for the suggestion. Table 2 has been added to summarize and illustrate the antiviral compounds used in this study (line 157).

3) Please, correct Remdesvir in schemes and figures (see Fig. 2 for example).

Response 3: Thank you for noticing this typo. The word has been corrected.

4) Page 15, line 509, Ref.63 (2021 - bold).

Response 4: The years of journal articles are bolded according to the journal guidelines.

Reviewer 4 Report

Comments and Suggestions for Authors

This is a good interesting paper but there are a few shortcomings.

1) There should be a table that summarises the compound used with a description of the modes of action e.g protease inhibitor, nucleoside analogue. Otherwise it is very confusing as it is.

2) There should be figures containing the chemical structures of the compound as this may indicate how the compounds bind to their targets.

3) Do you know the sequences in the RNA that nucleoside compounds are mimicking and if they are present in the canine distemper virus?

Author Response

The authors appreciate the comments and suggestions from reviewer 4

1) There should be a table that summarises the compound used with a description of the modes of action e.g protease inhibitor, nucleoside analogue. Otherwise it is very confusing as it is.

Response 1: Thank you for the suggestion. Table 2 has been added to summarize and illustrate the antiviral compounds used in this study (line 157).

2) There should be figures containing the chemical structures of the compound as this may indicate how the compounds bind to their targets.

Response 2: Thank you for the suggestion. Table 2 has been added to summarize and illustrate the antiviral compounds used in this study (line 157).

3) Do you know the sequences in the RNA that nucleoside compounds are mimicking and if they are present in the canine distemper virus?

Response 3: Nucleoside analogues in this study substitute the nucleosides themselves (nucleotides with no phosphate group). Remdesivir and GS-441524 are analogues of adenosine. EIDD2801 and EIDD1931 are analogues of cytosine.